# The Influence of Gender on Long-Term Cardiovascular Outcomes in Patients Undergoing Percutaneous Coronary Intervention for Acute Myocardial Infarction and the Association with Cardiac Left Ventricular Function

**DOI:** 10.3390/diagnostics15222901

**Published:** 2025-11-16

**Authors:** Vidar Ruddox, Ingvild Norum, Jøran Hjelmesæth, Thor Edvardsen, Jan Erik Otterstad

**Affiliations:** 1Department of Emergency Medicine, Vestfold Hospital Trust, 3103 Tønsberg, Norway; 2Department of Endocrinology, Obesity, and Nutrition, Vestfold Hospital Trust, 3103 Tønsberg, Norway; 3Department of Endocrinology, Morbid Obesity, and Preventive Medicine, Institute of Clinical Medicine, University of Oslo, 0313 Oslo, Norway; 4Department of Cardiology, Oslo University Hospital, Rikshospitalet, 0372 Oslo, Norway; 5Institute of Clinical Medicine, Faculty of Medicine, University of Oslo, 0313 Oslo, Norway; 6Department of Cardiology, Vestfold Hospital Trust, 3103 Tønsberg, Norway

**Keywords:** myocardial infarction, PCI, gender, major cardiovascular events

## Abstract

**Background/Objectives**: Traditionally, women have been observed to have older age, more co-morbidities, and poorer long-term clinical outcomes following acute myocardial infarction (AMI) when compared to men. However, age-adjusted analyses have demonstrated that gender differences are often attenuated, and the potential influence of left ventricular function and structure have been infrequently studied. The aim of the present study was to evaluate how LV function could influence gender differences in the long-term incidence of a composite of clinically relevant cardiovascular outcomes. **Methods**: Patients treated with early PCI for AMI were examined with echocardiography 2–4 days after the index AMI and followed by a mean 73 (±13) months. The primary endpoint was the incidence of a composite of total death, recurrent myocardial infarction, hospitalization for angina pectoris with an angiogram documenting progression of coronary artery stenoses, new heart failure, evidence of stroke/transient ischemic attack (TIA), and ventricular arrhythmia. **Results**: Among the 236 patients studied, 179 (76%) were men, with an average age of 66 (±11) years, and 57 were women (24%), with an age of 65 (±10) years. Men exhibited a higher incidence of anterior STEMI (*p* = 0.030), lower left ventricular ejection fraction (LVEF) (*p* = 0.02), reduced global longitudinal strain (*p* = 0.001), and larger left ventricular end-systolic volume index (LVESVI) (*p* = 0.007) compared to women. Both genders had similar peak troponin T values and symptom-to-needle times, as well as an equivalent number of stents implanted, prevalence of co-morbidities, and discharge medication. After sixyears of follow-up, Kaplan–Meier curves revealed better long-term cardiovascular outcome-free survival among women (log-rank *p* = 0.041). Cox regression analysis indicated that neither age nor LVEF influenced this gender difference, which, however, was reduced and became non-significant when LVESVI was added (HR 1.747 (95% CI 0.89–3.43)). No difference in mortality was observed, but men had significantly higher rates of heart failure (*p* = 0.03). **Conclusions**: This study demonstrated that men with a previous PCI-treated AMI had a two-fold (HR 2.155) higher risk of a composite long-term cardiovascular outcome as compared with women. The detrimental effect of male gender remained significant after adjustments for age and LVEF, but the male gender effect was reduced and became insignificant after adjustment for age and LVESVI. In view of this, our findings indicate that higher LVESVI may partly explain the detrimental effect of male gender on cardiovascular outcomes after PCI-treated AMI.

## 1. Introduction

In previous studies of unselected patients with acute myocardial infarction (AMI), women have consistently had a higher overall mortality rate, been of older age, and have been less likely to receive early invasive management and secondary prophylactic treatment compared to men [1,2,3]. However, the use of total mortality as an endpoint in the context of the modern treatment of AMI may be questionable, as cardiovascular deaths account for only approximately 50% of long-term fatalities [4,5,6]. Therefore, further investigation is needed to ascertain whether gender differences predict outcomes in a consecutive series of patients treated with early percutaneous coronary intervention (PCI) alongside aggressive lifestyle measures and medical therapies [7,8,9].

With women typically comprising only 25–30% of patients in post-AMI studies, it has been suggested that there may be a protective effect against AMI for women in comparison with men of a similar age [10]. Thus, it raises the question of whether men continue to have a higher risk than women following AMI. Most studies in this area have not included a comprehensive echocardiographic evaluation of left ventricular (LV) systolic function at enrollment. Murphy et al. [7] observed that female gender was independently associated with improved long-term survival following PCI treatment of ST-segment elevation AMI (STEMI) and that men had more significant LV dysfunction as assessed by LV ejection fraction, which was linked to worse long-term survival. However, the impact of potential differences in LV volumes was not reported. Additionally, White et al. [11] reported that LV end-systolic volume was the primary determinant of survival after recovery from AMI among men.

The aim of the present study, in view of the findings of White et al., was to evaluate how LV function may influence gender differences in the long-term incidence of a composite of cardiovascular outcomes in consecutive AMI patients treated with early PCI. Our hypothesis was that men, despite identical management to women, would have more pronounced LV systolic dysfunction, being associated with a less favorable clinical course.

## 2. Materials and Methods

This is a retrospective ancillary study of a recently published prospective observational study that examined the prognostic impact of changes in global longitudinal strain following PCI-treated AMI [12]. The present study incorporates a retrospective cohort design to compare long-term clinical cardiovascular outcomes between women and men, as well as a prospective cohort design involving interviews with patients about their lifestyle measures and medication after 5 years. In the former design, all data were verified from hospital records. In the latter, data on the present use of medication were based upon information obtained from patients during a structured telephone interview. The clinical follow-up was extended over 6 years to ensure a sufficient number of endpoints for analysis.

The primary endpoint was defined as a composite of total death, found by national registries, and the following, found by patient record screening: recurrent myocardial infarction, hospitalization for angina pectoris with an angiogram documenting progression of coronary artery stenoses, new heart failure, evidence of stroke/transient ischemic attack (TIA), and ventricular arrhythmia.

The study was conducted at Vestfold Hospital Trust, a secondary care general hospital. Eligible patients underwent PCI at Oslo University Hospital, a tertiary center. Following PCI, most patients returned to Vestfold Hospital Trust within 1–2 days for further management. Consecutive patients stabilized within 4 days following PCI between April 2016 and December 2018 were included in the study. After the index AMI, patients were to participate in a multidisciplinary cardiac rehabilitation program [13]; medical treatment was administrated according to current guidelines. Patients were followed from the time of the index AMI until February 2024 or death, with all available hospital records reviewed.

The study protocol was approved by the Regional Ethics Committee of Health Region South-East, Norway (Approval number 2015/2359). This ancillary extension of the original study involved a planned telephone interview after 5 years of follow-up. Informed consent was obtained for all subjects involved in the study.

Eligible patients had experienced AMI type 1 [14], including both STEMI and non-STEMI (NSTEMI). STEMIs and NSTEMIs were classified according to the most recent ESC guidelines at that time [14,15] and STEMI patients were further subdivided into anterior or non-anterior STEMI based on their electrocardiogram (ECG) findings upon admission. The coronary angiogram and subsequent PCI were performed according to prevailing guidelines at the time of study planning (2015) [16].

Endpoints except for death were adjudicated by a cardiologist and found by patient record screening. Recurrent MI was classified according to the most recent ESC guidelines at that time [15], angina pectoris was found by discharge diagnosis, and the coronary angiogram had to be performed according to current guidelines [16]. New heart failure, evidence of stroke/transient ischemic attack (TIA), and ventricular arrhythmia were found by discharge diagnosis.

The following exclusion criteria were applied [12]: hemodynamically unstable patients, ongoing atrial fibrillation/flutter/irregular heart rhythm, and life expectancy less than 2 years. It is important to note that there were no age or LV ejection fraction limits for inclusion.

During the initial echocardiographic examination all patients were weighed, and height was self-reported.

The specific echocardiography methods have been described in detail [12]. In short, volumes were measured with Simpson biplane measurements. Baseline measurements of global longitudinal strain, LV ejection fraction, and LV volume indexes were included as representative measures for LV systolic function. Additional measures included left atrial maximal volume index, septal e′ and lateral e′ velocities (cm/s), and E/e′ index. Valvular heart disease diagnoses were based on the presence of left-sided valvular regurgitations and stenoses classified as moderate or worse. Intraventricular thrombi were registered, treated with warfarin, and followed with repeated echocardiograms after 3–6 months.

Comorbidities were defined based on patients’ self-reports and information from hospital records. Hypertension was determined by treatment with at least one antihypertensive drug at inclusion. Baseline and new-onset diagnoses of cancer, diabetes mellitus, (HbA1c ≥ 6.5%) and/or treatment with antidiabetic drug), chronic inflammatory disease (such as systemic autoimmune disease, rheumatic disease, or inflammatory gastrointestinal disorders), and chronic obstructive pulmonary disease were also recorded.

The study methods involved a thorough assessment of long-term cardiovascular outcomes between inclusion and end-of-follow-up, according to available hospital records. Disagreements among investigators were resolved through consensus.

A telephone interview was conducted between March and October 2022 to collect additional variables including self-reported present weight, attempts at weight reduction, smoking status, level of physical activity, and present cardiovascular medication use including beta-blockers, statins, angiotensin converting enzyme (ACE) inhibitors, glucagon-like peptide (GLP) agonists, and platelet inhibitors or novel anticoagulants.

### Statistical Analyses

Descriptive statistics are reported as mean ± standard deviation or number of variables or events, when appropriate. The chi square test was applied to compare differences between categorical variables and the independent *t*-test to compare continuous variables. The primary endpoint, time to first cardiovascular outcome, was analyzed using the Kaplan–Meier method. The two gender groups were compared with the log rank test. Cox uni- and multivariate regression analyses were used for assessing the effect of gender on the incidence of long-term cardiovascular outcomes in models based on variables associated with the outcome in univariate analysis, clinical considerations, and correlation analysis.

Correlations between variables expressing systolic cardiac function, LVEF, LVEDVI, LVESVI, and GLS, were examined by bivariate Pearson correlation analysis. A secondary endpoint, the incidence of the 5 components of the composite outcome, was compared between genders using chi square analysis. Non-fatal endpoints were counted once, but if a patient experienced multiple different endpoints, all were counted.

A two-tailed *p* < 0.05 was considered significant for all statistical analyses. The analyses were performed using IBM SPSS version 29.0.0.

## 3. Results

A total of 289 patients were initially screened, and after excluding 53 patients due to unfeasible examination (two due to poor image quality), short life expectancy, or no consent to participate (Appendix A), 236 were included in the study. The patients were followed up for 73.4 (±12.7) months until death or February 2024. We experienced 27 deaths and one lost to follow-up during the study period. In addition, 208 patients were interviewed after a mean of 58.7 (±12.7) months.

Baseline characteristics are demonstrated in Table 1. The mean age of patients included was 64.9 (±10.2) years, with 24% being women and 76% being men. Men and women had a similar age and BMI, but men were on average 15.3 kg heavier than women. Previous coronary artery disease events, co-morbidities, and index AMI subtypes were similar between genders. Peak troponin T level and the number of stented vessels and stents implanted, as well as the time from symptom debut to PCI treatment for both STEMI and NSTEMI, were similar. The proportions of patients having a normal baseline LVEF equal to or above 50% for women and men were 74% and 55%, respectively (*p* < 0.05).

Men exhibited a higher incidence of anterior wall STEMI (*p* = 0.03), had lower LV ejection fraction (*p* = 0.02) and global longitudinal strain (*p* = 0.001), larger LVESVI (*p* = 0.007), and maximal left atrial volume index compared to women. Tissue Doppler variables were similar between genders. Mean LVESVI among men was 36% higher than in women.

The 55 patients with anterior STEMI (23%) were slightly younger, had lower LV ejection fraction and global longitudinal strain (*p* < 0.001 for both), and larger LVESVI (*p* < 0.005) than the 181 patients with non-anterior STEMI and NSTEMI (Appendix A).

At discharge, all patients were on double platelet inhibition. The use of beta-blockers, statins, and ACE inhibitors and the participation rates in a cardiac rehabilitation program were similar in both genders.

The lifestyle measures and medications after five years follow-up are demonstrated in Table 2. At that time, 51 women and 157 men were available for an adequate telephone interview. Men still weighed 15.3 kg more than women. Men had higher physical activity levels and a higher prevalence of statin use compared to women. More women than men were still smoking. There was no gender difference in patients using intensive statin therapy, beta-blockers, ACE inhibitors, or GLP agonists.

During the follow-up period, a first long-term cardiovascular outcome occurred in 86 patients, 18.6% women (*n* = 16) and 31.9% men (*n* = 70) (HR 2.155 (95% CI 1.134–4.093)). The incidence of the first long-term cardiovascular outcome was not significantly different between patients with anterior STEMI (28.4%) and non-anterior STEMI (31.6%).

Long-term cardiovascular outcome-free survival curves are shown in Figure 1, presenting a worse outcome among men compared with women (*p* = 0.041).

A univariate analysis of all variables associated with the primary outcome is presented in Table 3 and in a bivariate Pearson correlation analysis (Appendix A). The correlation analysis showed that LVESVI correlated with all other indices of left ventricular function, both systolic (i.e., LVEF, LVEDVI, GLS) and diastolic (E/e′ and LAVI), and thus LVESVI was chosen to be the most clinically relevant and statistically compatible variable concerning systolic LV function. Based on this, we included the variables of gender, age, and LVESVI in a multivariate Cox regression analysis. As LVEF is an established marker in echocardiography, LVEF was included in a fourth model.

In the Cox multivariate regression analysis (Table 4), gender was still a significant predictor of long-term cardiovascular outcome after adjustment for age (Model 2). However, the estimated effect (HR) of gender on the composite primary outcome was reduced and became non-significant when LVESVI was added to the model (Model 3). By contrast, adding LVEF to gender and age (Model 4) did not influence the estimated effect of gender on long-term cardiovascular outcome.

Secondary endpoints are demonstrated in Table 5. A total of 27 (11.4%) patients died, with 13 (48%) of the deaths categorized as cardiovascular, with no differences between men and women. All heart failure events occurred in men (*p* = 0.03), and men had a threefold higher incidence of recurrent AMI compared with women. The incidences of angina pectoris and stroke were similar between genders. None experienced ventricular arrhythmias causing hospitalization.

**Table 5 diagnostics-15-02901-t005:** Secondary endpoints: Mortality data and incidence of non-fatal endpoints counted as at least one per patient. All data presented as *n* (%).

Event	Women, *n* = 57	Men, *n* = 179	*p*-Value
**Fatal**			
Total mortality	8 (14.0%)	19 (10.6%)	n.s.
Cardiovascular mortality	3 (5.3%)	10 (5.6%)	n.s.
**Non-fatal**			
Recurrent myocardial infarction	3 (5.3%)	25 (15.0%)	n.s.
Hospitalization for angina	4 (7.0%)	20 (11.2%)	n.s.
Heart failure	0	15 (8.4%)	*p* < 0.05
Stroke/TIA	4 (7.0%)	11 (6.1%)	n.s.
Total non-fatal events	11 (19.3%)	53 (29.6%)	n.s.

## 4. Discussion

In the present study, we observed notable differences between men and women in their clinical characteristics following an AMI. Specifically, men were heavier, had a higher incidence of anterior wall STEMI, and had a poorer LV systolic function. On the other hand, women demonstrated a better long-term cardiovascular outcome-free survival as evidenced by the Kaplan–Meier curves and log rank analysis. However, it is worth noting that LVESVI emerged statistically as a significant predictor for long-term cardiovascular outcomes, thus mitigating the influence of gender.

One notable strength of our study, although a recall bias may have been present, is the long and practically complete follow-up period, during which all reported events were verified from hospitalization records and patient contacts. Moreover, as optimized post–myocardial infarction medical therapies in reducing arrhythmic events are closely proportional to left ventricular ejection fraction (LVEF) [17] and recent evidence highlights their contribution to lowering the incidence of sudden cardiac death in post-MI patients [18], secondary prophylaxis by large is evenly distributed between groups.

Additionally, our findings of a significantly larger LVESVI among men in a previous study of normal men and women aged 30–60 years using identical two-dimensional echocardiographic measurements [19] further support the notion that these observed differences in the post-AMI population may not be derived from a physiological pattern. The mean gender difference in LVESVI in that study was only 21%, i.e., far less than in the present post-AMI population. Furthermore, men in the present study had a higher incidence of heart failure, indicating an unfavorable impact on their clinical outcomes, despite similar peak troponin T, number of stents implanted, and co-morbidities in both genders at inclusion.

In view of the small study population, we expanded the follow-up time for events from five (time of telephone interview) to six years. In that time, a primary endpoint was registered in more than one third of our patients. This pattern of a small population with a relatively high number of events had an impact on the statistical methods used. We chose to let univariate analysis guide the construction of the multivariate Cox model to an extent, e.g., troponin T was omitted as HR and 95% CI since both were 1.00. We included the most significant echocardiographic measurement reflecting systolic function LVESVI, in addition to LVEF, as the latter is the most widely used prognostic predictor in echocardiography.

Despite the better prognosis for women, men had higher physical activity levels and a higher prevalence of statin use compared to women and more women than men were still smoking.

Interestingly, our findings challenge the historical assumption that women are protected by their estrogen prior to menopause, resulting in later onset of AMI with worse clinical outcome. In fact, a study by Mehili et al. in 2002 [20] found similar age-adjusted one-year mortality rates among men and women after AMI predominately treated with PCI. However, unlike our study, women in that study were significantly older and had a higher incidence of anterior AMI than men.

A more recent study found that women compared to men had a significantly higher incidence of all-cause death, recurrent AMI, and hospitalization for heart failure at follow-up (203 (33.4%) vs. 428 (26.9%); *p* = 0.002) with age being an independent predictor [21].

Other studies have reported mixed findings regarding gender differences in outcomes following AMI. For instance, Josiah and Farshid [22] found a significantly higher unadjusted one-year mortality among women compared to men in a retrospective study. However, on multivariate analysis, female sex did not emerge as an independent predictor of death. Conversely, studies in France [8] and Australia [7] have shown that women have a lower five-year mortality and improved long-term survival rates, respectively, following PCI. In another study involving patients undergoing PCI for acute coronary syndrome, women had a higher incidence of one-year MACE, but these differences were attenuated after multivariate adjustment [23]. Furthermore, a large registry study from Sweden and the UK [9] found that women experienced higher one-year all-cause mortality after PCI for coronary artery disease, highlighting the impact of age at the time of PCI as a strong predictor of mortality in this population. All these studies lack information on the predictive value of LV volumes.

A sub-study of the Valiant trial showed that men have larger LV volumes than women and similar LV ejection fraction [24], with the larger LV volumes in men being associated with a greater risk of death and composite cardiovascular endpoints. However, this study included patients with evidence of heart failure post-AMI and only one-third of them underwent PCI.

Additionally, a study conducted in the UK [25] found that higher LV volumes were associated with increased mortality and composite cardiovascular endpoints in men, but not in women, suggesting that sex differences in LV remodeling may influence mortality risk in the general population. However, this study included individuals of older age and may not be applicable to younger post-AMI patients treated with PCI. These findings, along with ours and those of White et al. [11], are hypothesis-generating and need to be corroborated in studies of apparently normal individuals of younger age and post-AMI patients treated with PCI.

A striking feature of our relatively small study is the small age difference between the genders, which did not represent any selection bias since we had no age limits or different inclusion criteria for women versus men. This might, however, be due to chance. The larger LVESVI and its association with long-term cardiovascular outcomes among men cannot be explained by their higher incidence of anterior STEMI. During follow-up, women fared better despite potential handicaps, such as a higher percentage of current smokers, less use of statins, and lower levels of exercise. Possible “protecting factors” might include estrogen, although most included women had been through menopause.

We are unaware of studies reporting a significant treatment effect on outcomes associated with a reduction in LVESVI among post-AMI patients treated with early PCI, where most have been discharged with a more or less preserved LV ejection fraction. In our study, there was no significant gender difference in long-term treatment with beta-blockers or ACE inhibitors. In this context, the difference in association between larger LV volumes and death in the Valiant study [23] was not influenced by the use of renin–angiotensin inhibitors, which was similar in men and women.

### Limitations

The major limitation of this study is the small number of patients, especially women, and the potential for type 2 errors cannot be excluded. We do suggest that LVESVI may mitigate sex differences in outcomes and, while plausible, it could also reflect different LV remodeling pathways rather than a direct causal chain. We did not include any diastolic indices as 40% of the total population had reduced systolic function which in turn affects diastolic function. With that reservation in mind, tissue Doppler variables reflecting LV relaxation were similar in both genders, as was a crude estimation of LV end-diastolic pressure as derived from the E/e′ ratio. Moreover, men had larger left atrial volumes, but this may have been associated with their larger LVESVI and do not necessarily reflect diastolic dysfunction. Furthermore, the structured telephone interview may be hampered by a recall bias. Another limitation is the lack of data for age at menopause or estrogen substitution among women.

## 5. Conclusions

This study demonstrated that men with a previous PCI-treated AMI had a two-fold (HR 2.155) higher risk of a composite long-term cardiovascular outcome as compared with women. The detrimental effect of male gender remained significant after adjustments for age and LVEF, but the male gender effect was reduced and became insignificant after adjustment for age and LVESVI. In view of this, our findings indicate that higher LVESVI may partly explain the detrimental effect of male gender on cardiovascular outcomes after PCI-treated AMI.

## Figures and Tables

**Figure 1 diagnostics-15-02901-f001:**
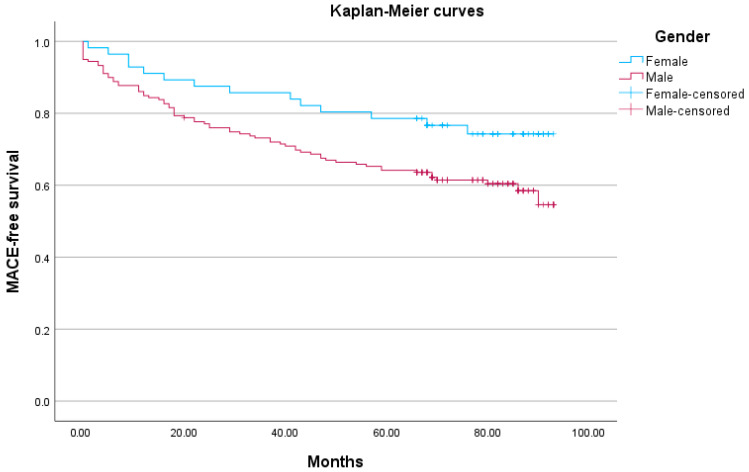
Kaplan–Meier curves for time to first long-term cardiovascular outcome (Log rank test *p* = 0.041).

**Table 1 diagnostics-15-02901-t001:** Gender differences among 236 patients included. Unless otherwise indicated, data are presented as *n* (%).

Variables	Women, *n* = 57	Men, *n* = 179	*p*-Value
**Characteristics**			
Age, years, mean (SD)	65.8 (11.3)	64.6 (9.9)	n.s.
BMI, mean (SD)	26.3 (5.2)	27.4 (3.9)	n.s.
Weight, kg, mean (SD)	72.4 (15.4)	87.7 (13.6)	*p* < 0.001
Feeling of stress before index MI	25 (43.9%)	76 (43.6%)	n.s.
**Previous coronary artery disease**			
AMI	8 (14.0%)	35 (19.6%)	n.s.
PCI	7 (12.3%)	36 (20.1%)	n.s.
CABG	2 (3.5%)	12 (6.7%)	n.s.
**Co-morbidities**			
Hypertension	26 (45.6%)	80 (44.7%)	n.s.
Diabetes mellitus	14 (24.6%)	34 (19.0%)	n.s.
Cancer	5 (8.8%)	12 (6.9%)	n.s.
COPD	5 (8.8%)	13 (7.4%)	n.s.
Chronic inflammatory disorders	4 (7.0%)	10 (5.7%)	n.s.
Current smoker	17 (29.8%)	43 (24.0%)	n.s.
**AMI subtypes and PCI procedures**			
STEMI	26 (45.6%)	102 (57%)	n.s.
Anterior wall STEMI	7 (12.3%)	48 (26.8%)	*p* < 0.05
Troponin T peak, ng/L median (IQR)	598 (2208)	1644 (3876)	n.s.
Time to PCI, STEMI, h median (IQR)	3.5 (4.0)	3.5 (3.0)	n.s.
Time to PCI, NSTEMI, h median (IQR)	48 (36)	48 (42)	n.s.
Arteries stented	1.44 (0.76)	1.63 (0.78)	n.s.
Stents implanted	2.00 (1.30)	2.02 (1.30)	n.s.
**Echocardiographic variables, mean (SD)**			
LV ejection fraction, %	52.0 (6.1)	49.5 (8.1)	*p* < 0.05
LVEDVI, mL/m^2^, mean	67.5 (14.2)	87.23 (19.0)	n.s.
LVESVI, mL/m^2^, mean	33.1 (9.5)	45.2 (15.0)	*p* < 0.01
Global longitudinal strain, %	−15.5 (3.5)	−14.0 (3.3)	*p* < 0.01
Maximal LAVI, mL/m^2^	28.5 (9.7)	33.0 (9.4)	*p* < 0.01
Septal e′, cm/s	6.0 (1.9)	6.4 (1.8)	n.s.
Lateral e′, cm/s	7.3 (1.9)	7.2 (2.4)	n.s.
E/e′ ratio	10.5 (3.4)	10.5 (3.1)	n.s.
**Associated valvular disease**			
Total	1 (1.8%)	13 (7.3%)	*p* < 0.001
**Cardioprotective medication at discharge**			
Double platelet inhibition	57 (100%)	179 (100%)	n.s.
Betablocker	45 (78.9%)	131 (73.2%)	n.s.
Betablocker dose, mg mean (SD) ^	42.8 (37.0)	44.9 (39.9)	n.s.
Statin	57 (100%)	176 (89.0%)	n.s.
ACE inhibitor	18 (31.6%)	76 (42.5%)	n.s.
**Cardiac rehabilitation**			
Participants	36 (65.5%)	126 (70.4%)	n.s.

^ Doses equivalent for metoprolol depot in mg. Abbreviations: BMI = body mass index; COPD = chronic obstructive pulmonary disease; CABG = coronary artery bypass graft; h = hours; LV = left ventricular; LVEDVI = Left ventricular end-diatolic volume index; LVESVI = Left ventricular end-systolic volume index; LAVI = left atrial volume index. Global longitudinal strain expressed as positive values.

**Table 2 diagnostics-15-02901-t002:** Clinical data obtained from the telephone interview conducted between February and October 2022 from adequate contact with 208 patients at that time. Data are presented as in Table 1.

Variables at the Time of Interview	Women, *n* = 51	Men, *n* = 157	*p*-Value
**Lifestyle**			
Weight, kg mean (SD)	70.6 (15.4)	86.7 (15.1)	*p* < 0.001
Intentional weight loss	19 (37.0%)	71 (45.2%)	n.s.
Attended weight loss program	3 (5.9%)	7 (4.5%)	n.s.
Physical activity			*p* < 0.01 (Overall difference)
Grade 1	12 (23.5%)	14 (9.0%)	
Grade 2	8 (15.7%)	15 (9.6%)	
Grade 3	31 (60.8%)	127 (81.4%)	
Still smokers	14 (25.0%)	20 (11.6%)	*p* < 0.05
**Cardiovascular medication**			
All were on a platelet inhibitor or a novel anticoagulant *			
Betablocker	29 (56%)	92 (58.6%)	n.s.
Betablocker dose, mg mean (SD)	59.2 (44.8%)	64.3 (47.6%)	n.s.
Statin	44 (86.3%)	153 (97.5%)	*p* < 0.05
Intense statin treatment	37 (72.5%)	138 (87.9%)	n.s.
ACE inhibitor	17 (33.3%)	55 (35.0%)	n.s.
GLP agonist	3 (5.9%)	3 (1.9%)	n.s.

* Predominantly apixaban.

**Table 3 diagnostics-15-02901-t003:** A univariate analysis of all variables associated with the primary outcome. Data are presented as hazard ratios with 95% confidence intervals.

Variable	Hazard Ratio (95% CI)
Gender	2.155 (1.134–4.093)
Age	1.027 (1.003–1.052)
BMI	1.007 (0.955–1.062)
Troponin T peak level	1.000 (1.000–1.000)
Time to PCI	1.006 (1.000–1.011)
N arteries	1.374 (1.046–1.805)
N stents	1.110 (0.957–1.287)
LVEF	0.970 (0.941–0.999)
LVEDVI	1.018 (1.007–1.030)
LVESVI	1.026 (1.012–1.041)
GLS	0.915 (0.854–0.981)
Max LAVI	1.036 (1.012–1.061)
E/e′	1.090 (1.073–1.162)

Abbreviations: BMI = body mass index; PCI = percutaneous coronary intervention; LVEF = left ventricular ejection fraction; LVEDVI = left ventricular end-diastolic volume index; LVESVI = left ventricular end-systolic volume index; GLS = global longitudinal strain; LAVI = left atrium volume index.

**Table 4 diagnostics-15-02901-t004:** Cox regression analysis (numbered 1–4) of the influence of age and clinically and statistically relevant variables on gender difference in long-term cardiovascular outcome.

	Hazard Ratio (95% CI)	*p*-Value
Model 1		
Gender	2.155 (1.134–4.093)	0.019
Model 2		
Gender	2.334 (1.224–4.452)	0.010
Age	1.032 (1.007–1.057)	0.012
Model 3		
Gender	1.747 (0.889–3.434)	0.105
Age	1.038 (1.013–1.064)	0.002
LVESVI (mL/m^2^)	1.028 (1.011–1.044)	<0.001
Model: Age, LVESVI (mL/m^2^)		
Model 4		
Gender	2.263 (1.181–4.338)	0.014
Age	1.036 (1.011–1.061)	0.005
LVEF	0.967 (0.939–0.997)	0.030

Abbreviations: LVESVI = left ventricular end-systolic volume index; LVEF = left ventricular ejection fraction.

## Data Availability

The data presented in this study are available on request from the corresponding author. The data are not publicly available due to regulatory reasons.

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
