# Peer review of "The Influence of Gender on Long-Term Cardiovascular Outcomes in Patients Undergoing Percutaneous Coronary Intervention for Acute Myocardial Infarction and the Association with Cardiac Left Ventricular Function"

_diagnostics, 2025, doi:10.3390/diagnostics15222901_

Round 1

Reviewer 1 Report (Previous Reviewer 1)

Comments and Suggestions for Authors

In the discussion on cardiovascular outcomes, the authors should at least briefly mention of the role of optimized post–myocardial infarction medical therapy in reducing arrhythmic events, which are closely proportional to left ventricular ejection fraction (LVEF) (10.3390/jcm10081618). Moreover, recent evidence highlights its contribution to lowering the incidence of sudden cardiac death in post-MI patients (10.1016/j.hrthm.2025.09.023).

Author Response

Comment1: In the discussion on cardiovascular outcomes, the authors should at least briefly mention of the role of optimized post–myocardial infarction medical therapy in reducing arrhythmic events, which are closely proportional to left ventricular ejection fraction (LVEF)

Response1: We agree and therefore have included the two references and the sentence: “optimized post–myocardial infarction medical therapy in reducing arrhythmic events are closely proportional to left ventricular ejection fraction (LVEF) [17] and recent evidence highlights its contribution to lowering the incidence of sudden cardiac death in post-MI patients [18], secondary prophylaxis by large is evenly distributed between groups” to the manuscript.

Reviewer 2 Report (New Reviewer)

Comments and Suggestions for Authors

This manuscript evaluates gender differences in long-term cardiovascular outcomes after PCI-treated AMI, with a particular focus on LV systolic function and LV volumes. The study addresses an important and persistent clinical question: whether sex-related prognosis differences in contemporary AMI care remain after accounting for cardiac function.

The analysis is thoughtfully designed, the follow-up is impressively long, and the endpoint adjudication appears thorough. The finding that LV end-systolic volume largely attenuates the observed gender effect is clinically relevant and hypothesis-generating.

However, some methodological and reporting aspects require clarification or expansion before publication.

Major Comments:

  1. Given the use of a composite primary endpoint, it would be helpful for the authors to clearly define each component event and briefly justify their inclusion. The current composite combines outcomes with different clinical weights (e.g., death, heart failure, recurrent ischemia), and providing explicit definitions and adjudication criteria would clarify event classification and strengthen interpretability of the results.
  2. Given the central role of LV volumes in the authors’ conclusions, the echocardiographic methodology requires clearer description. It would be helpful to specify how volumes were measured (e.g., biplane Simpson), whether any studies were excluded for inadequate image quality, and if reproducibility metrics were available.
  3. Please consider to include the number at risk below the Kaplan–Meier curves in Figure 1.
  4. In the discussion, when referring to large registries demonstrating sex-related differences in post-PCI outcomes, it may be helpful to also consider more recent evidence examining sex and age interactions in contemporary AMI cohorts, including patients with both obstructive and non-obstructive presentations. Studies such as the recent European registry analysis (cite PMID: 37261384) evaluating sex- and age-related outcome differences in MINOCA vs MIOCA populations provide further context on the multifactorial contributors to sex disparities in ischemic heart disease and could be briefly acknowledged to enrich the discussion and emphasize the evolving understanding in this field.
  5. The Discussion suggests LVESVI may “explain” sex differences in outcome. While plausible, this could also reflect LV remodeling pathways rather than a direct causal chain; please temper causal language and more clearly acknowledge confounding and biological hypotheses.
  6. Given that lifestyle factors and medication adherence were assessed through patient interview several years after the index PCI, the manuscript would benefit from a clearer discussion of potential recall bias.

Minor Comments:

  1. Please report median and IQR for skewed variables (troponin, follow-up time).
  2. Clarify whether STEMI anteriority was included or tested in multivariable models.
  3. Consider reporting GLS as negative values (conventional format) or clearly state directionality in all figures/tables.
  4. Some typographical issues (e.g., “vakues”, formatting of Table 1 smokers %, “Corcuøation”) require revision.

Author Response

We sincerely thank reviewer 2 for encouraging and insightful comments. Our response will be given for each comment separately:

Comment1: Given the use of a composite primary endpoint, it would be helpful for the authors to clearly define each component event and briefly justify their inclusion. The current composite combines outcomes with different clinical weights (e.g., death, heart failure, recurrent ischemia), and providing explicit definitions and adjudication criteria would clarify event classification and strengthen interpretability of the results.

Response1: We agree that the separate endpoints and adjudication criteria is not clearly stated. We have rewritten the following passage: “The primary endpoint was defined as a composite of total death found by national registries, and the following found by patient record screening; recurrent myocardial infarction, hospitalization for angina pectoris with an angiogram documenting progression of coronary artery stenoses, new heart failure, evidence of stroke/transient ischemic attack (TIA) and ventricular arrhythmia”, and added the following: “Endpoints except for death were adjudicated by a cardiologist and found by patient record screening. Recurrent MI was classified according to the most recent ESC guidelines at that time [16], angina pectoris was found by discharge diagnosis and the coronary angiogram had to be performed according to current guidelines [15]. New heart failure, evidence of stroke/transient ischemic attack (TIA) and ventricular arrhythmia was found by discharge diagnosis”.

Comment2: Given the central role of LV volumes in the authors’ conclusions, the echocardiographic methodology requires clearer description. It would be helpful to specify how volumes were measured (e.g., biplane Simpson), whether any studies were excluded for inadequate image quality, and if reproducibility metrics were available.

Response2: We agree that this has to be clarified and thank you for this comment. We have added In short, volumes were measured with Simpson biplane measurements. to the methods section and the following to the first sentence in the results section: (two due to poor image quality). A detailed flow chart is provided in the supplemental material at the end of the manuscript. We do have reproducibility metrics available for performance at our center, but have omitted referring to this study in the present manuscript as we feel that it is outside the scope of the article: Otterstad JE, Froeland G, St. John Sutton M, Holme I. Accuracy and reproducibility of biplane two-dimensional chocardiographic measurements of left ventricular dimension and function. Eur Heart J 1997; 18: 507-13.

Comment3: In the discussion, when referring to large registries demonstrating sex-related differences in post-PCI outcomes, it may be helpful to also consider more recent evidence examining sex and age interactions in contemporary AMI cohorts, including patients with both obstructive and non-obstructive presentations. Studies such as the recent European registry analysis (cite PMID: 37261384) evaluating sex- and age-related outcome differences in MINOCA vs MIOCA populations provide further context on the multifactorial contributors to sex disparities in ischemic heart disease and could be briefly acknowledged to enrich the discussion and emphasize the evolving understanding in this field.

Response3: Thank you very much for bringing this very interesting article to our attention. We have included the following in the discussion: A more recent study found that women compared to men had a significantly higher incidence of all-cause death, recurrent AMI, and hospitalization for heart failure at follow-up (203 (33.4%) vs. 428 (26.9%); P = 0.002) with age being an independent predictor [21].

Comment4: The Discussion suggests LVESVI may “explain” sex differences in outcome. While plausible, this could also reflect LV remodeling pathways rather than a direct causal chain; please temper causal language and more clearly acknowledge confounding and biological hypotheses.

Response4: We agree that our findings do not suggest causality and that the discussion should reflect this. We have emphasized that the effect is statistically significant and in the limitations section we have included the following sentence: We do suggest that LVESVI may mitigate sex differences in outcome and while plausible, it could also reflect different LV remodeling pathways rather than a direct causal chain.

Comment5: Given that lifestyle factors and medication adherence were assessed through patient interview several years after the index PCI, the manuscript would benefit from a clearer discussion of potential recall bias.

Response5: We agree and have included this in the second para in the discussion: although a recall bias may have been present as well as pointing It out in the limitations.

Minor comments have been addressed.

Round 2

Reviewer 1 Report (Previous Reviewer 1)

Comments and Suggestions for Authors

Very good improvements. No other comments 

Reviewer 2 Report (New Reviewer)

Comments and Suggestions for Authors

Thank you to the authors for the revisions made, which I believe have enhanced the quality of the final manuscript. I have no further comments.

This manuscript is a resubmission of an earlier submission. The following is a list of the peer review reports and author responses from that submission.

Round 1

Reviewer 1 Report

Comments and Suggestions for Authors

Major Comments

  • The introduction sets up the hypothesis that men will have more pronounced LV systolic dysfunction despite identical management. However, this hypothesis appears partially contradicted by later Kaplan–Meier findings showing better survival in women but with LVESVI potentially explaining the difference. It would help to make the causal pathway clearer from the outset, including how LVESVI fits into the conceptual model
  • The combination of retrospective and prospective components (retrospective cohort for outcomes and prospective follow-up interviews) could introduce different biases. The manuscript would benefit from a dedicated section in Materials and Methods clarifying how these designs were integrated and how possible recall bias from interviews was handled. Please specify it also in the limitations section.
  • The discussion omits the potential impact of post-MI arrhythmias on long-term outcomes (10.4103/2221-6189.336578) and the possible role of mechanical circulatory support in patients with severe LV dysfunction (10.3390/jcm10081618). Even without data in this cohort, a brief mention would enhance the quality and relevance of the manuscript.
  • All tables should be reformatted to improve readability, with clearer alignment, consistent units, and sufficient spacing for easier interpretation.

Minor comment

  • The manuscript would benefit from minor English language polishing for smoother readability (e.g., “fared better off” could be revised to “fared better”).
  • Define all abbreviations at first use in the main text (e.g., MACE, GLP agonists) even if they are common in cardiology literature.
  • The manuscript sometimes alternates between “LVESVI” and “LVSEVI” — standardize the term throughout.
  • Reference 14, citing the third universal definition of myocardial infarction, should be updated to the most recent available definition to ensure alignment with current clinical standards.
  • The abstract contains a formatting issue: the "Background" section appears bold and visually separated from the rest, which is presented as distinct text blocks. The layout should be standardized for consistency and readability.
Comments on the Quality of English Language

see below

Author Response

Comment 1. The introduction sets up the hypothesis that men will have more pronounced LV systolic dysfunction despite identical management. However, this hypothesis appears partially contradicted by later Kaplan–Meier findings showing better survival in women but with LVESVI potentially explaining the difference. It would help to make the causal pathway clearer from the outset, including how LVESVI fits into the conceptual model

Response 1. We are thankful as you are absolutely correct; We have included Harvey D Whites findings from reference 11 and added: «Additionally, White et al [11] reported that LV end systolic volume was the primary determinant of survival after recovery from AMI among men.” in the introduction.

Comment 2. The combination of retrospective and prospective components (retrospective cohort for outcomes and prospective follow-up interviews) could introduce different biases. The manuscript would benefit from a dedicated section in Materials and Methods clarifying how these designs were integrated and how possible recall bias from interviews was handled. Please specify it also in the limitations section.

Response 2. This was a topic of discussion in the planning of this ancillary study. To clarify, we have added the following in the Methods section: “In the former design, all data were verified from hospital records. In the latter, data on present use of medication, was based upon information obtained from patients during a structured telephone interview.”

To the Discussion/limitations we added: “Furthermore, the structured telephone interview may be hampered by a recall bias.”

Comment 3. The discussion omits the potential impact of post-MI arrhythmias on long-term outcomes (10.4103/2221-6189.336578) and the possible role of mechanical circulatory support in patients with severe LV dysfunction (10.3390/jcm10081618). Even without data in this cohort, a brief mention would enhance the quality and relevance of the manuscript.

Response 3. Thank you for pointing this out. MACE did in fact include ventricular arrhythmias. Accordingly we have updated Methods to this end. None of our patients experienced ventricular arrhythmias causing hospitalization and that is the reason for us forgetting to list this very important outcome as a part of MACE.

Minor comments: All minor comments have been corrected to the best of our abilities.

Reviewer 2 Report

Comments and Suggestions for Authors

The manuscript entitled "The influence of gender on long-term cardiovascular outcomes in patients undergoing percutaneous coronary intervention for acute myocardial infarction and the association with cardiac left ventricular function" addresses an interesting topic. The authors investigate sex differences in long-term major adverse cardiovascular events (MACE) after PCI for AMI, with a particular focus on left ventricular end-systolic volume index (LVESVI) as a potential mediator. However, several areas require attention to strengthen the manuscript. 

First, the abstract would benefit from restructuring according to the standard scientific format. The authors should clearly separate background, methods, results, and conclusions. 

Second, the manuscript should include a dedicated limitations section near the end of the discussion. 

Third, while the focus on LVESVI as a mediator is interesting, causality should be presented cautiously. The language in the discussion should be adjusted to reflect that the observed association does not prove mediation. 

Finally, the manuscript would benefit from a central illustration.

Several minor revisions would also improve clarity: 
-P-values should be reported in a consistent format. 
-The manuscript should clarify whether troponin T values represent peak or admission levels. -Including the proportion of patients on guideline-directed medical therapy at follow-up would help contextualize the findings.

In summary, this is a potentially impactful manuscript. I recommend major revision before consideration for publication.

Author Response

Comment 1. First, the abstract would benefit from restructuring according to the standard scientific format. The authors should clearly separate background, methods, results, and conclusions. 

Response 1. Agreed. We have reformatted the abstract, as well as made a few improvements and clarifications as requested from the reviewers.

Comment 2. Second, the manuscript should include a dedicated limitations section near the end of the discussion. 

Response 2. Agreed. This has been done.

Comment 3. Third, while the focus on LVESVI as a mediator is interesting, causality should be presented cautiously. The language in the discussion should be adjusted to reflect that the observed association does not prove mediation.

Response 3. Thank you for an important observation. We did have discussions prior to submission on this topic exactly. We have now edited both the discussion and conclusion (and the abstract accordlingly). The text “possible role as a mediator for” is changed to “association with" in the discussion. The conclusion has been rephrased to “In the present study, LVESVI is found to be a predictor for the higher incidence of MACE among men after PCI treated AMI. »

Minor revisions has been corrected. P-values are reported in a consistent format and we have clarified that the troponin T values are at peak.

Reviewer 3 Report

Comments and Suggestions for Authors

I thoroughly read your article and found it quite interesting especially given the lack of studies addressing gender differences, with most existing literature focusing primarily on male-presenting diseases.

First things first: When I initially glanced at your article, the tables stood out as problematic. They lack borders and structure, making them difficult to follow. Additionally, there was an entire blank page between tables, which disrupts the flow. I strongly recommend thorough formatting and editing as the first step.

  1. Line 130: Could you clarify which specific anticoagulant was included in your study?
  2. Lines 156–157: It appears you performed propensity matching between male and female participants, but this isn’t mentioned in the Methods section. If this is the case, please include it there.
  3. Line 162: Please provide the p-value for LAVI.
  4. Lines 171–176 and 177–179: Kindly add p-values for the differences mentioned in these sections.
  5. To improve the readability of your tables, consider adding a third column specifically for p-values.

Author Response

Comment 1. When I initially glanced at your article, the tables stood out as problematic. They lack borders and structure, making them difficult to follow. Additionally, there was an entire blank page between tables, which disrupts the flow. I strongly recommend thorough formatting and editing as the first step.

Response 1. We are sorry for the trouble all reviewers went throug reading our manuscript. We hope this is fully amended in the resubmitted manuscript.

Comment 2. Could you clarify which specific anticoagulant was included in your study? 

Response 2. We do not have the exact number, but all were DOACs, predominantly Apixaban.

Comment 3. It appears you performed propensity matching between male and female participants, but this isn’t mentioned in the Methods section. If this is the case, please include it there.

Response 3. We did in fact not perform propensity score matching.

Comment 4. p-values

Response 4. p-values are now reported consistently thoughout the manuscript and tables are now properly formatted.

Round 2

Reviewer 1 Report

Comments and Suggestions for Authors

Thank you for your reply to my review. However, your response to major comment n°3 does not address my comment. The authors should include a citation of the following article 10.3390/jcm10081618 when they discuss the different outcomes in HF. This work should be added at line 246, after “a significant role in the observed outcome.” Please make sure this reference is properly cited in the revised manuscript.

Author Response

Comment 1. The authors should include a citation of the following article 10.3390/jcm10081618 when they discuss the different outcomes in HF.

Response 1. This citation has been included as reference 17 in the manuscript. Inclusion of this reference was not intentionally left out in the first revision and we thank the reviewer for reminding us.

Reviewer 2 Report

Comments and Suggestions for Authors

Thank you for your thorough revisions and for addressing all the comments provided. The manuscript is suitable for publication in its current form.

Author Response

Comment 1. Thank you for your thorough revisions and for addressing all the comments provided. The manuscript is suitable for publication in its current form.

Response 1. We sincerely thank reviewer 2 for help in improving the mansucript.